# Novel Dairy Fermentates Have Differential Effects on Key Immune Responses Associated with Viral Immunity and Inflammation in Dendritic Cells

**DOI:** 10.3390/foods13152392

**Published:** 2024-07-29

**Authors:** Dearbhla Finnegan, Claire Connolly, Monica A. Mechoud, Jamie A. FitzGerald, Tom Beresford, Harsh Mathur, Lorraine Brennan, Paul D. Cotter, Christine E. Loscher

**Affiliations:** 1School of Biotechnology, Dublin City University, D09 DX63 Dublin, Ireland; dearbhla.finnegan@dcu.ie; 2Food for Health Ireland, Science Centre South (S2.79), University College Dublin, Dublin 4, Ireland; claire.connolly.2@ucdconnect.ie (C.C.); monica.mechoud@teagasc.ie (M.A.M.); jamie.fitzgerald@ucd.ie (J.A.F.); tom.beresford@teagasc.ie (T.B.); harsh.mathur@teagasc.ie (H.M.); lorraine.brennan@ucd.ie (L.B.); paul.cotter@teagasc.ie (P.D.C.); 3UCD School of Agriculture and Food Science, University College Dublin, D04V1W8 Dublin, Ireland; 4Teagasc Food Research Centre, Moorepark, Fermoy, P61 C996 Co. Cork, Ireland; 5APC Microbiome Ireland, Biosciences Institute, Biosciences Research Institute, University College Cork, T12 R229 Cork, Ireland; 6VistaMilk, P61 C996 Co. Cork, Ireland

**Keywords:** fermentates, functional food, immune boosting, immunomodulation, anti-inflammatory, viral immunity, chronic inflammation

## Abstract

Fermented foods and ingredients, including furmenties derived from lactic acid bacteria (LAB) in dairy products, can modulate the immune system. Here, we describe the use of reconstituted skimmed milk powder to generate novel fermentates from *Lactobacillus helveticus* strains SC232, SC234, SC212, and SC210, and from *Lacticaseibacillus casei* strains SC209 and SC229, and demonstrate, using in vitro assays, that these fermentates can differentially modulate cytokine secretion via bone-marrow-derived dendritic cells (BMDCs) when activated with either the viral ligand loxoribine or an inflammatory stimulus, lipopolysaccharide. Specifically, we demonstrate that SC232 and SC234 increase cytokines IL-6, TNF-α, IL-12p40, IL-23, IL-27, and IL-10 and decrease IL-1β in primary bone-marrow-derived dendritic cells (BMDCs) stimulated with a viral ligand. In contrast, exposure of these cells to SC212 and SC210 resulted in increased IL-10, IL-1β, IL-23, and decreased IL-12p40 following activation of the cells with the inflammatory stimulus LPS. Interestingly, SC209 and SC229 had little or no effect on cytokine secretion by BMDCs. Overall, our data demonstrate that these novel fermentates have specific effects and can differentially enhance key immune mechanisms that are critical to viral immune responses, or can suppress responses involved in chronic inflammatory conditions, such as ulcerative colitis (UC), and Crohn’s disease (CD).

## 1. Introduction

In recent years, there has been a clear shift in the interest of consumers in food for health, with people more than ever before being aware of the ancient saying “Let food be thy medicine and medicine be thy food”. Food can be a critical contributor to a healthy, disease-free quality of life. The corollary is also true in that global dietary risk factors are estimated to cause 11 million deaths and amount to 255 million disability-adjusted life years annually [1]. It is a long-known fact that what we eat influences our body, and vital nutrients are essential for growth, cellular function, tissue development, energy, and immune defence [2]. There is growing evidence of the role of specific foods and food components as immunomodulators, improving immune defence function and increasing resistance to infection while maintaining tolerance [3]. Furthermore, deficiencies in key nutrients can result in the development of disease, putting an individual at greater risk of disease development or susceptibility to viral infections.

The term “fermentates” generally refers to “a powdered preparation, derived from a fermented [food] product and which can contain the fermenting microorganisms, components of these microorganisms, culture supernatants, fermented substrates, and a range of metabolites and bioactive components” [4]. Our previous work outlined the potential of fermentates as immune-boosting functional food ingredients for supporting the immune system in the defence against viral invasion in macrophage models [5]. Furthermore, the popularity of and demand for functional food components as natural immune fitness boosters are growing within the industry [6]. This demonstrates the commercial desire for functional food ingredients, such as fermentates, to boost the immune system and support immune fitness. Indeed, with the recent increases in viral outbreaks, including the SARS-CoV-2 virus, the Monkeypox virus, and the Langya virus, as well as the yearly outbreaks of seasonal influenza, there is heightened global interest in maintaining and enhancing viral immunity [5,6]. Second to our previous study into the effects of fermentates on viral immunity in macrophages [5], this study is one of the first to explore the effects of novel fermentates from *Lb. helveticus* SC234, SC232, SC212, and SC210 and *Lacticaseibacillus casei* SC229 and SC209 on immune mechanisms that are critical to viral immunity, as well as those critical to anti-inflammatory immune responses. 

The discovery of novel anti-inflammatory food ingredients is important given that the modern diet consists of greater meat and animal product consumption, and observational studies have linked such dietary patterns to the risk of developing inflammatory bowel diseases (IBDs), such as Crohn’s disease (CD) and ulcerative colitis (UC) [7]. These chronic digestive diseases affect over 10 million people worldwide and have no known cause or cure [8]. IBDs are issues within the body associated with gut-related symptoms as a result of immunological imbalances within the intestinal mucosa associated with cells of the adaptive immune system [9]. IBDs arise as a result of the immune system responding inappropriately and triggering chronic inflammation. Dietary approaches to mitigate the effects of IBDs include enteral nutrition in the form of hydrolysed or intact protein formulas; however, there is much interest in identifying more palatable and easier-to-use dietary approaches [7]. Otherwise, nonsurgical strategies involve medications to control symptoms, including aminosalicylates, antibiotics, biologics, and corticosteroids, which often have significant side effects [10,11]. The prevalence of IBDs is on the rise [12,13], and thus finding a functional food that could aid in the control and management of such prevalent and debilitating conditions, in a natural way, would be of huge medical significance.

We used murine dendritic cells challenged with either a viral immune stimulus, namely loxoribine (LOX), or an inflammatory immune stimulus lipopolysaccharide (LPS; *Escherichia coli* 055:B5). LOX is a TLR2 ligand and mimics the immune response triggered by a viral infection, while LPS is a TLR4 ligand and mimics the immune response seen in chronic inflammation, in diseases such as UC and CD. The effects of these fermentates on cell viability and cytokine secretion for the quantification of interleukin (IL)-1β, IL-6, IL-10, tumour necrosis factor (TNF)-α, IL-12p40, IL-23, and IL-27 were investigated in Toll-like receptor ligand (TLR)-activated dendritic cells in order to establish the specificity or differential effects of these fermentates on these cells. In this study, we examine the effects of a range of fermentates (SC232, SC234, SC212, SC210, SC209, and SC229) on dendritic cell function in the context of viral immunity as well as chronic inflammation, such as that associated with inflammatory bowel diseases.

## 2. Materials and Methods

### 2.1. Generation of Reconstituted Skimmed Milk (RSM)-Based Fermentates

Skimmed milk powder (SMP) was used as a substrate for the generation of the fermentates used in this study. The SMP was reconstituted at 10% *w*/*v* in distilled water, autoclaved at 121 °C for 5 min, cooled, and stored at 4 °C for a maximum of 7 days. Working cultures were prepared for each strain from the respective frozen mother culture stocks of strains *Lb. helveticus* SC232, *Lb. helveticus* SC234, *Lacticaseibacillus casei* SC229, *Lb. casei* SC209, *Lb. helveticus* SC210, and *Lb. helveticus* SC212 (all previously prepared in 10% *w*/*v* RSM), and incubated for 24 h at 37 °C under aerobic conditions without agitation. From these cultures, a further inoculum was added to the 10% *w*/*v* RSM and incubated for 24 h at 37 °C under aerobic conditions again without agitation. Fermentates were then heat-treated to inactivate the respective LAB strain for 20 min at 72 °C. After cooling to room temperature, the pH of the fermentates was neutralised. These fermentates were aliquoted and immediately frozen at −80 °C until further analysis. Non-fermented RSM samples subjected to the same heat-treatment procedure mentioned above were used as negative controls for all experiments described herein. The codes SC232, SC229, SC209, SC210, SC212, and SC234 will hereafter be used when referring to the individual fermentates produced.

### 2.2. Cell Culture

Bone-marrow-derived dendritic cells (BMDCs), harvested from the bone marrow of a female BALB/c mice of 6–8 weeks old obtained from Charles River (Margate, UK), were cultured in complete medium RPMI-1640 (Biosciences, Dublin, Ireland) containing 5 ng/mL GM-CSF (Merck, Haverhill, UK) and supplemented with 10% FCS and 1% Pen-Strep. Cells were cultured for 7 days at 37 °C, with 5% CO_2_ and 95% humidified air. Cell seeding for experimentation was carried out at a concentration of 1 × 10^6^ cells/mL. Cells were left overnight to settle before proceeding with experimentation.

### 2.3. Cell Viability

Cell viability was determined using the CellTiter 96^®^ AQueous One Solution Cell Proliferation Assay and conducted as per the manufacturer’s instructions (MyBio, Kilkenny, Ireland). Dendritic cells were seeded at a concentration of 1 × 10^6^ cells/mL in a flat-bottom 96-well plate, and incubated for 24 h at 37 °C in an atmosphere of 95% humidified air and 5% CO_2_. Cells were treated with 25 mg/mL (based on fermentates containing 10% protein on a weight basis) of the fermentate for 1 h and incubated under the same conditions, before stimulation with LOX 0.5 mM and LPS 100 ng/mL [14,15,16] for 24 h. Dimethylsulfoxide (DMSO) was included as a positive control to induce cell death. After 24 h, 20 μL of the thawed CellTiter96^®^ Aqueous One Solution was added to each well of the 96-well plate and incubated at 37 °C for 3 h in a humidified 5% CO_2_ atmosphere. Absorbance was read at 490 nm using a Versamax^TM^ 96-well plate reader (VWR, Dublin, Ireland). Cell viability was expressed as the percentage viability of the treatment group relative to the control group. 

### 2.4. Enzyme-Linked Immunosorbent Assays (ELISA)

Determination, via sandwich ELISAs, of the effects of the fermentate samples on cytokine production in the activated dendritic cells required harvesting of the cell supernatants and subsequent analysis using commercial DuoSet ELISA kits (R&D Systems Europe, Abdingdon, Oxon, UK) according to the manufacturer’s instructions. This allowed for the quantification of the cytokines IL-1β (range: 15.6–1000 pg/mL), IL-6 (range: 15.6–1000 pg/mL), IL-10 (range: 31.2–2000 pg/mL), TNF-α (range: 31.2–2000 pg/mL), IL-12p40 (range: 31.2–2000 pg/mL), IL-12p70 (range: 39.1–2500 pg/mL), IL-23 (range: 39.1–2500 pg/mL), and IL-27(range: 15.6–1000 pg/mL).

### 2.5. Metabolomics

Metabolomics analysis of the fermentate samples was performed using a nuclear magnetic resonance (^1^H-NMR) spectroscopy approach (N = 54). Following centrifugation at 13,500× *g* for 15 min at 4 °C, the samples were filtered using 3 kDa ultra-centrifugal filters and the sample filtrate was frozen at −80 °C until further analysis (Sigma-Aldrich, Merck KGaA, Darmstadt, Germany). On the day of analysis, the resulting filtrate was combined with 10 μL sodium trimethyl silyl [2,2,3,3-2H4] propionate (TSP) and 200 μL deuterium oxide (D_2_O). Spectra were acquired with a 600 MHz Varian Spectrometer (Varian Limited, Oxford, UK). Spectra were acquired using 16,384 complex points and 256 scans. All ^1^H-NMR spectra were referenced to TSP at 0.0 parts per million (ppm) and processed manually with the Chenomx NMR Suite (version 7.7) using a line broadening of 0.2 Hz, followed by phase and baseline correction. 

### 2.6. Statistical Analysis

Statistical analysis was carried out using GraphPad Prism software v.10, using a one-way ANOVA to compare variance among the means of different sample groups. A Newman–Keuls post hoc test was used to determine significance among the samples. The level of statistical significance was indicated by * (*p* < 0.05), ** (*p* < 0.01), and *** (*p* < 0.001). Statistical testing for significant differences was carried out (*n* = 3) with respect to the non-fermented RSM samples, which were used as negative controls for all experiments described herein. Experiments were repeated three times and the results shown are representative of all three experiments.

### 2.7. Ethical Statement

The care, treatment, and experiments involved in this study were approved by the Research Ethics Committee (REC) of Dublin City University (Approval ID: DCUREC/2023/187). Animals were housed in DCU’s biological resource unit in specific pathogen-free environments until culling via cervical dislocation for cell and organ harvesting for use in vitro. Animals were monitored daily for their health and welfare and documented accordingly.

## 3. Results

### 3.1. Cell Viability Is Not Affected by the Presence of Fermentate Samples

Initially, preliminary studies carried out by the laboratory on a large panel of fermentates established that leaving the fermentate raw and unprocessed was optimal for fermentate bioactivity, as opposed to pretreatment in the form of centrifugation, filtering, or centrifugation and filtering combined. Therefore, to conduct further in-depth analysis, the raw unprocessed fermentate was used for the assessment of fermentates in DCs. An MTS assay was performed to determine whether fermentates SC232, SC234, SC212, SC210, SC209, or SC229, in the presence/absence of LOX or LPS, had any significant effect on BMDC viability. Figure 1A–C demonstrate that there was no significant change in cell viability following the exposure of BMDCs to LOX or LPS in the presence of SC232 or SC234 Lb. helveticus strains (Figure 1A), SC212 or SC210 Lb. helveticus strains (Figure 1B), or SC209 or SC229 Lacticaseibacillus casei strains (Figure 1C). The use of the positive control of 10% DMSO provided a clear demonstration of the cytotoxic effects on cell viability.

### 3.2. The Effects of Fermentates on Cytokine Secretion

An ELISA was then performed in accordance with manufacturer’s instructions in order to assess the effects of our novel fermentates in primary dendritic cells in the absence/presence of LOX or LPS. See the Appendix A for the confidence intervals applied.

Figure 2 shows that our novel fermentates SC232 and SC234 significantly affected the secretion of cytokines in response to immune challenges with LOX and LPS when compared to the respective control/TLRs in BMDCs. IL-1β (*p* < 0.001), IL-6 (*p* < 0.001), IL-12p40 (*p* < 0.002), and TNF-α (*p* < 0.033) were significantly increased in the presence of LOX, with only low levels of IL-10, IL-23, and IL-27, and no IL-12p70 detectable. IL-6, IL-12p40, and IL-27 were significantly increased in the presence of LPS (*p* < 0.001), as well as TNF-α (*p* < 0.002), with only low levels of IL-1β, IL-10, and IL-23, and similarly no IL-12p70. The concentrations of IL-12p70 secreted were so low that it was not considered statistically significant. 

In the presence of LOX, SC232 increased IL-6 (*p* < 0.001), TNF-α (*p* < 0.001), IL-12p40 (*p* < 0.001), IL-27 (*p* < 0.001), IL-10 (*p* < 0.002), and IL-23 (*p* < 0.002). In contrast, SC234 increased IL-10 (*p* < 0.001), TNF-α (*p* < 0.001), and IL-6 (*p* < 0.002), but decreased IL-1β (*p* < 0.001) in the presence of LOX. In the presence of LPS, SC232 increased TNF-α (*p* < 0.001), but decreased IL-12p40 (*p* < 0.002). SC234 increased IL-10 (*p* < 0.001), TNF-α (*p* < 0.001), and IL-6 (*p* < 0.002), but decreased IL-12p40 (*p* < 0.002) and IL-27 (*p* < 0.002) in the presence of LPS. Interestingly, in the absence of TLR stimulation, SC234 increased IL-10 (*p* < 0.033), TNF-α (*p* < 0.033), and IL-6 (*p* < 0.002).

Figure 3 demonstrates that fermentates SC212 and SC210 significantly affected the secretion of cytokines in response to immune challenges with LOX and LPS when compared to their respective controls/TLRs in BMDCs. 

In the presence of LOX, SC212 increased IL-10 (*p* < 0.001) and IL-6 (*p* < 0.002), but decreased IL-1β (*p* < 0.001) and IL-12p40 (*p* < 0.001), while having no significant effects on the other cytokines measured when compared to the control cells. SC210 increased IL-10 (*p* < 0.001) and IL-6 (*p* < 0.033) but decreased IL-1β (*p* < 0.001) and IL-12p40 (*p* < 0.001) in the presence of LOX, with no significant effects on the other cytokines. In the presence of LPS, SC212 increased IL-10 (*p* < 0.001) and IL-23 (*p* < 0.001), but decreased IL-12p40 (*p* < 0.033). SC210 increased IL-1β (*p* < 0.002), IL-10 (*p* < 0.001), and IL-23 (*p* < 0.001) in the presence of LPS. Interestingly, in the absence of TLR stimulation, SC212 alone increased IL-1β (*p* < 0.002), IL-6 (*p* < 0.001), IL-10 (*p* < 0.001), TNF-α (*p* < 0.001), IL-23 (*p* < 0.002), and IL-27 (*p* < 0.001). Similarly, SC210 in the absence of TLR stimulation increased IL-1β (*p* < 0.002), IL-6 (*p* < 0.001), IL-10 (*p* < 0.001), TNF-α (*p* < 0.001), IL-23 (*p* < 0.001), and IL-27 (*p* < 0.001), but, in addition, it also increased IL-12p40 (*p* < 0.033). 

Figure 4 demonstrates that fermentates SC209 and SC229 significantly affected the secretion of cytokines in response to immune challenges with LOX and LPS when compared to their respective controls/TLRs in BMDCs. 

In the presence of LOX, SC209 increased IL-12p40 (*p* < 0.001), IL-6 (*p* < 0.002), and IL-27 (*p* < 0.002), but decreased IL-1β (*p* < 0.001). SC229 increased IL-12p40 (*p* < 0.001) and IL-6 (*p* < 0.002) in the presence of LOX. In the presence of LPS, SC209 increased IL-1β (*p* < 0.002). In contrast, SC229 increased IL-12p40 (*p* < 0.001) in the presence of LPS but had no significant effect on the other cytokines. Interestingly, in the absence of TLR stimulation, SC229 increased IL-1β (*p* < 0.001), TNF-α (*p* < 0.002), and IL-12p40 (*p* < 0.033). In contrast, SC209 had no significant effect on any cytokines secreted in the absence of TLR stimulation. 

We also examined the ratio of IL-12 to IL-10. Overall, SC212 and SC210 had a much greater anti-inflammatory effect than SC232 or SC234, which was evident in a greater change in the IL-12/IL-10 ratio. For example, the IL-12p40/IL-10 expression ratio for the cells alone was 38:1, which decreased to 4:1 in the presence of SC212, while the IL-12p40/IL-10 expression ratio for the cells alone versus that for the cells with SC232 only decreased from 244:1 to 219:1, respectively. The IL-12p40/IL-10 expression ratio for the LOX-activated cells alone was 36:1, which decreased to 6:1 for the LOX-activated cells in the presence of SC212, while the IL-12p40/IL-10 expression ratio for the LOX-activated cells alone was 19:1 and this decreased to 17:1 for the LOX-activated cells in the presence of SC232. Similarly, in the LPS-activated cells, there was a much larger anti-inflammatory effect in the presence of SC212 compared to the LPS-activated cells alone, as the expression ratio was decreased from 90:1 to 4:1, versus SC232, which decreased this ratio from 43:1 to 8:1. 

Table 1 highlights the differing immune-boosting and anti-inflammatory effects on cytokine secretion observed in the presence of SC232, SC234, SC212, and SC210. Although there was some crossover in their profiling, we can see that SC232 and SC234 had more immune-boosting effects in the LOX-activated cells, with increased levels of IL-6, TNF-α, IL-23, and IL-27, and less immune-boosting effects in the LPS-activated cells, with only slightly increased TNF-α, IL-1β, and IL-27 levels and a decreased level of IL-12p40. In contrast, SC212 and SC210 had greater anti-inflammatory capabilities in the LPS-activated cells, showing the highest secretion of IL-10 and a decreased secretion of IL-12p40, with lesser anti-inflammatory abilities in the LOX-activated cells. This demonstrates the context-dependant specificity of the fermentates.

### 3.3. Metabolite Levels in the Fermentates

Given our data, which demonstrate differential effects of the fermentates on cytokine production, we carried out a metabolomics analysis to determine whether there were also differences between the metabolite levels of the fermentates. In total, 46 metabolites were analysed, but only desaminotyrosine, succinate, glutamate, and lactate showed any significant differences in their metabolite levels and are therefore highlighted below. Results for the remaining metabolites are included in the Appendix A. Figure 5 demonstrates that metabolite levels varied in the skimmed milk powder fermentate, with higher levels of metabolites such as desaminotyrosine, succinate, glutamate, and lactate in SC212. Interestingly, glutamate was also elevated in SC218, while succinate was also elevated in SC232. 

## 4. Discussion

We have previously demonstrated a role for novel fermentates SC232 and SC234 as potential immune-boosting supplements through their ability to enhance key macrophage functions, including cytokine secretion, chemokine secretion, nitric oxide production, phagocytosis, and cell surface marker expression [5]. In this study, we aimed to expand this work to assess their effects in another type of key immune cell, dendritic cells, and to compare these effects to a range of other novel fermentates. Dendritic cells are considered the master regulators of the immune response and play critical roles in antigen presentation by capturing, processing, and presenting antigens to lymphocytes in order to initiate and regulate the adaptive immune response [17]. They play a critical role in viral immunity and chronic inflammation [18,19]; therefore, any effect on these cells imposed by fermentates could indicate their ability to have significant influence on shaping immune responses. Importantly, BMDCs are critical in maintaining immunological homeostasis through controlling innate and adaptive immune responses via migration of these cells to inflamed sites in IBDs like CD and UC, as well as to the lymph nodes, to aid in resolving inflammation [20]. The TLR7 ligand, LOX, was used in order to mimic an immune response to a viral antigen, while the TLR4 ligand, LPS, was used in order to mimic an inflammatory immune response, similar to that seen in chronic inflammatory disorders such as UC and CD. Assessing fermentates’ effects in LOX- and LPS-activated dendritic cells has enabled us to determine any specificity that fermentates may have. According to previous research conducted in our laboratory [5], preliminary studies carried out on a large panel of fermentates used a dose range of 5 mg/mL, 10 mg/mL, 25 mg/mL, and 50 mg/mL fermentates and revealed 25 mg/mL as the optimal dose for fermentate bioactivity. Therefore, a dose of 25 mg/mL was used for this further in-depth analysis.

We demonstrate that cytokine secretion in BMDCs is differentially influenced by the presence of fermentates and that these effects differ depending on the mode of activation of the cell. This study demonstrates the specificity of SC232 and SC234 as immune-boosting fermentates in the context of a viral infection, and SC212 and SC210 as anti-inflammatory fermentates. Furthermore, SC209 and SC229 had little or no effects on dendritic cells’ cytokine secretion and this emphasises the unique specificity of the fermentates. 

When activated with the viral ligand LOX in the presence of SC232 and SC234, the BMDCs showed enhanced levels of secretion of key viral cytokines IL-6, TNF-α, IL-12p40, IL-23, and IL-27, as well as IL-10, which supports viral clearance. These effects were not the same in the presence of LPS, where the BMDCs showed decreased IL-12p40 and IL-27, which further supports the fermentates’ possible specificity in enhancing viral immunity. Given the importance of IL-6, TNF-α, IL-12p40, IL-23, and IL-27 in aiding the immune system during viral infection and supporting viral immunity, a fermentate that can enhance these cytokines could indeed be beneficial. 

It is well established that viral infections such as COVID-19 are characterised by increases in pro-inflammatory cytokines such as IL-6, TNF-α, IL-12p40, and IL-23 [21,22], while influenza and HIV-1 infections are characterised by increased production of IL-1β, IL-6, and TNF-α [23,24]. These cytokines are released by the immune system in an attempt to prevent or resolve the infection caused by the invading virus. IL-6, TNF-α, IL-12p40, and IL-23 are pro-inflammatory cytokines with key immunomodulatory and antiviral roles, including supporting and promoting CD4+ and CD8+ cell differentiation, B cell activation, additional cytokine secretion, the inhibition of viral replication, leukocyte trafficking, immune complex clearance, and the defence of intracellular organisms against invading viruses and pathogens [25,26,27,28,29,30,31,32]. When increased to a small degree, IL-10 plays a supportive role in viral clearance through regulation of the adaptive immune response and IL-10-secreting T cells [33,34]. 

Previous research has shown that fermentation products as well as food-derived compounds can influence cytokines associated with the viral immune response. Specifically, Kawashima et al. reported that in a randomised, double-blind, placebo-controlled clinical trial, *Pediococcus acidilactici* K15 was found to increase levels of both IL-6 and IL-10, ultimately increasing sIgA concentrations at mucosal sites in humans, aiding in host defence against pathogens and maintaining symbiosis with microorganisms present in the small intestine [34]. Furthermore, other studies by Takeda et al. showed that LAB, in particular the strain *Lactiplantibacillus plantarum* 06CC2, from cow’s cheese, increased the production of IL-12 and IL-12p40 in vitro and in vivo [35]. *Lb. plantarum* 06CC2 has been associated with the enhancement of the Th1 response, and has been found to result in the alleviation of influenza virus infection in mice [36]. In a recent study by Zhu et al., jasmine, found in the Chinese herbal medicine of jasmine tea, was found to exhibit antiviral activity in vitro [37] and increased production of TNF-α from RAW 264.7 cells occurred at higher concentrations of jasmine, ultimately demonstrating antiviral activity against HSV-1, shown via a plaque reduction assay [37]. *Panax ginseng* Meyer, or white ginseng, has the ability to increase IL-1β, IL-6, TNF-α, and IL-23, via activation of the MAPK kinase (MKK)4-c-Jun N-terminal kinase (JNK)-c-Jun signalling pathway [38,39]. This is important considering the fact that viral infections like that of the influenza virus decrease innate IL-23 and IL-12p70 concentrations and ultimately decrease IL-17 and IFN-γ responses, making the host more susceptible to further infection [40]. Therefore, our data showing that SC232 and SC234 can specifically modulate these cytokines could suggest their application in supporting viral immunity. 

The effects of SC212 and SC210 were different to those seen with the other fermentates. When activated with LOX and LPS in the presence of SC212 and SC210, the BMDCs showed enhanced levels of IL-10 and decreased secretion of pro-inflammatory cytokines IL-12p40 and IL-1β. 

It is well established that increased pro-inflammatory cytokine secretion, including increased levels of IL-1β, IL-12p40, IL-6, TNF-α, and IL-27, has been linked to chronic inflammation and disease progression, particularly in the cases of UC and CD [41,42,43,44,45,46,47,48]. An over-activation of the immune response produces these increased levels of pro-inflammatory cytokines, resulting in chronic inflammation and disease severity. IL-1β, IL-12p40, IL-6, TNF-α, and IL-27 are pro-inflammatory cytokines with key immunomodulatory effects and contribute to the chronic inflammation seen in UC and CD. They have functions in modulating immune responses, immune activation, host resistance, and immune-mediated inflammation—specifically, they play roles in blocking Treg and Th17 differentiation, restricting IL-2 production, promoting Th1 cells, amplifying CD8 responders that inhibit Th2 mucosal responses, leucocyte trafficking, immune complex clearance, the activation of endothelial cells, B and T cell proliferation and differentiation, Ig production, T cell activation and differentiation, natural killer cell stimulation, cytokine production, and activating acute phase responses [26,28,29,30,31,32,49,50,51,52,53,54,55,56,57,58,59,60,61,62,63,64]. When increased to a larger degree, IL-10 plays an important immunoregulatory role in the resolution of inflammation and inhibition of disease pathogenesis. IL-10 functions through increasing CD4-produced Th2 cytokine responses; suppresses Th1 responses; downregulates the antigen presentation capacities of antigen-presenting cells (APCs); inhibits the activation and effector function of T cells, monocytes, and macrophages; limits host immune response to invading pathogens; and ultimately prevents damage to the host from the over-activation of pro-inflammatory molecules [33,65,66,67]. 

Previous research has demonstrated that modulation of these cytokines can have anti-inflammatory effects. In a mouse model of UC and CD, compound LASSBio-1524 and its three analogues LASSBio-1760, LASSBio- 1763, and LASSBio-1764 reduced the secretion of TNF-α, IL-1β, IL-6, IL-12, and IFN-γ and increased the secretion of IL-10, leading to a reduction in measures of disease severity, including colonic tissue damage, the infiltration of inflammatory cells, and the production and expression of pro-inflammatory mediators, suggesting its role as a novel therapeutic for the treatment of UC and CD [68]. *Saccharomyces boulardii*, a widely used gastrointestinal treatment, has been shown to decrease levels of IL-1β, IL-6, and TNF-α in order to aid in suppressing colonic inflammation in a DSS-induced murine model of UC [69]. In a rat model of colitis, curcumin was shown to decrease disease activity, with reduced colonic mucosa damage, through the decreased expression of IL-27 via inhibition of the TLR4/NF-κB signalling pathway [70]. Furthermore, curcumin has also been shown to improve UC symptoms and alleviate chronic inflammation in DSS-induced colitis mouse models via reducing pro-inflammatory cytokines IL-1β, IL-2, IL-6, IL-9, IL-17A, IL-27, TNF-α, and C-C motif chemokine ligand 2 (CCL2), and promoting the anti-inflammatory cytokine IL-10 in colonic tissue [71,72,73,74]. Thus, it is clear from the literature that reducing the secretion of pro-inflammatory cytokines such as IL-1β, IL-6, IL-12, TNF-α, and IL-27 while increasing the secretion of anti-inflammatory cytokine IL-10 can be linked to an improved disease prognosis for UC and CD. Therefore, fermentates SC212 and SC210 could be novel candidates for the management of UC and CD in humans due to their ability to reduce the secretion of pro-inflammatory cytokines such as IL-1β, IL-6, IL-12, TNF-α, and IL-27 while increasing the production of anti-inflammatory cytokine IL-10. 

Our data demonstrating the differential effects of our fermentates on both viral immunity and inflammation are further supported by our additional findings that SC209 and SC229 had little or no bioactivity when subjected to the same immune stimuli, LOX and LPS. Therefore, not all fermentates influence cytokine secretion in dendritic cells and those that do have an influence differ in their effects, demonstrating that fermentates have unique specificity in their bioactivity, as opposed to there being a generalised bioactivity as a result of fermentate presence. 

Given the differences in the bioactivity of the fermentates, we examined whether there were any differences in the metabolites present. We chose SC232, as an immune-boosting fermentate, and SC212, as a key anti-inflammatory fermentate, as well as additional fermentates SC218, SC40, and SC215 from our database for contrast. The data clearly indicated differential metabolite profiles between the fermentates. 

SC232 has been clearly identified as an immune-boosting fermentate, while SC212 has a mixed profile. SC232 shows a higher concentration of succinate, as does SC212, while SC212 also has a higher concentration of desaminotyrosine. Studies showing increased levels of desaminotyrosine suggest a role for this metabolite in viral immunity and gut health, thus suggesting a role for SC212 as an antiviral and an anti-inflammatory due to its high levels of desaminotyrosine in comparison to its RSM. Desaminotyrosine influences mucosal barrier function; is important in maintaining mucosal immune homeostasis, protecting barrier integrity; has been found to attenuate DSS-induced mucosal inflammation; protects mice from bacterial endotoxin-induced septic shock; and is associated with enhanced viral immunity, while also having anti-inflammatory effects and effects on influenza, as an induction of the human-associated gut microbe *Clostridium orbiscindens* occurs to produce desaminotyrosine [75,76,77,78]. Succinate is an immunomodulator that regulates the function of immune cells in the intestine, increasing the abundance of tight-junction proteins claudin-1, zona occluden (ZO)-1, and ZO-2, to aid in intestinal epithelial barrier function and improving mucosal barrier integrity and microbial dysbiosis for improved gut health [79,80,81]. Succinate, as well as suppressing immune responses, can support inflammation and promote immune-boosting mechanisms through promoting the expression of inflammatory cytokines interleukin (IL)-25, IL-10, IL-8, and IL-18, which aids in antiviral ability and reduces viral infections, including vesicular stomatitis virus and influenza virus infections [82,83]. However, in recent reviews, succinate has been elevated in patients with IBDs, suggesting the role of high levels of succinate in IBD-associated mucosal inflammation and its potential detrimental effects to the gut microbiome [84,85]. 

SC212 has been identified as a largely anti-inflammatory fermentate. This is evidenced by the fact that SC212, as well as SC218, shows significantly higher levels of glutamate and lactate. It is most noteworthy that SC212 has the highest levels of glutamate and lactate, because, as demonstrated in the literature, these are anti-inflammatory metabolites largely associated with gut health, and thus a high presence of such metabolites may aid in gut health and the alleviation of IBD-associated inflammation. Lactate is a distinct signalling molecule that acts as a metabolic feedback regulator and regulates cells, receptors, mediators, and microenvironment-specific effects that augment Th17 cell, macrophage (specifically M2), tumour-associated macrophage, and neutrophil functions, all with significant effects in cancer, sepsis, autoimmunity, and wound healing [86]. 

Therefore, not only does fermentate SC232 have the ability to increase pro-inflammatory cytokines related to enhanced viral immunity, it also contains a higher concentration of antiviral-associated metabolites. In contrast, SC212 has the ability to enhance the secretion of anti-inflammatory cytokines to aid in mitigating the chronic inflammation associated with IBDs, while also containing higher levels of metabolites associated with anti-inflammatory effects and improved gut health. Similarly, we hypothesise that SC234, with its ability to enhance pro-inflammatory cytokine secretion for viral immunity, may also have higher concentrations of the antiviral-associated metabolites desaminotyrosine and succinate. On the other hand, SC210, with its anti-inflammatory cytokine secretion, may have higher concentrations of the anti-inflammatory-associated metabolites glutamate and lactate. Overall, however, while the above-mentioned metabolites may have specific roles in the fermentates described here, further studies are required to elucidate and confirm the precise mechanisms of action of these fermentates from a metabolomics point of view; however, this is beyond the scope of this current study reported here.

## 5. Conclusions

As demonstrated in the current literature available on similar functional foods, and building on previous work from our research group, we suggest a role for fermentates SC232 and SC234 as potential novel candidates for defence against viral infection in humans. This is due to their ability to support the secretion of pro-inflammatory cytokines IL-6, TNF-α, and IL-27 while increasing the production of anti-inflammatory cytokine IL-10, maintaining immune homeostasis, preventing viral persistence, and, ultimately, preventing viral infection, as well as demonstrating high levels of viral-associated metabolites such as desaminotyrosine and succinate. Furthermore, novel fermentates SC212 and SC210 have been highlighted as potential novel anti-inflammatory candidates that may be useful in the control, management, and treatment of the chronic inflammation that is often seen in UC and CD. This is due to their ability to support the secretion of anti-inflammatory cytokine IL-10 while decreasing pro-inflammatory cytokines IL-1β and IL-12p40, which is beneficial for an enhanced immune response and the resolution of chronic inflammation, as well as their demonstration of high levels of anti-inflammatory gut-health-associated metabolites such as glutamate and lactate. Bioactivity is unique and specific to different fermentates when challenged with different immune stimuli, thus suggesting their specific roles in modulating the immune system. 

## Figures and Tables

**Figure 1 foods-13-02392-f001:**
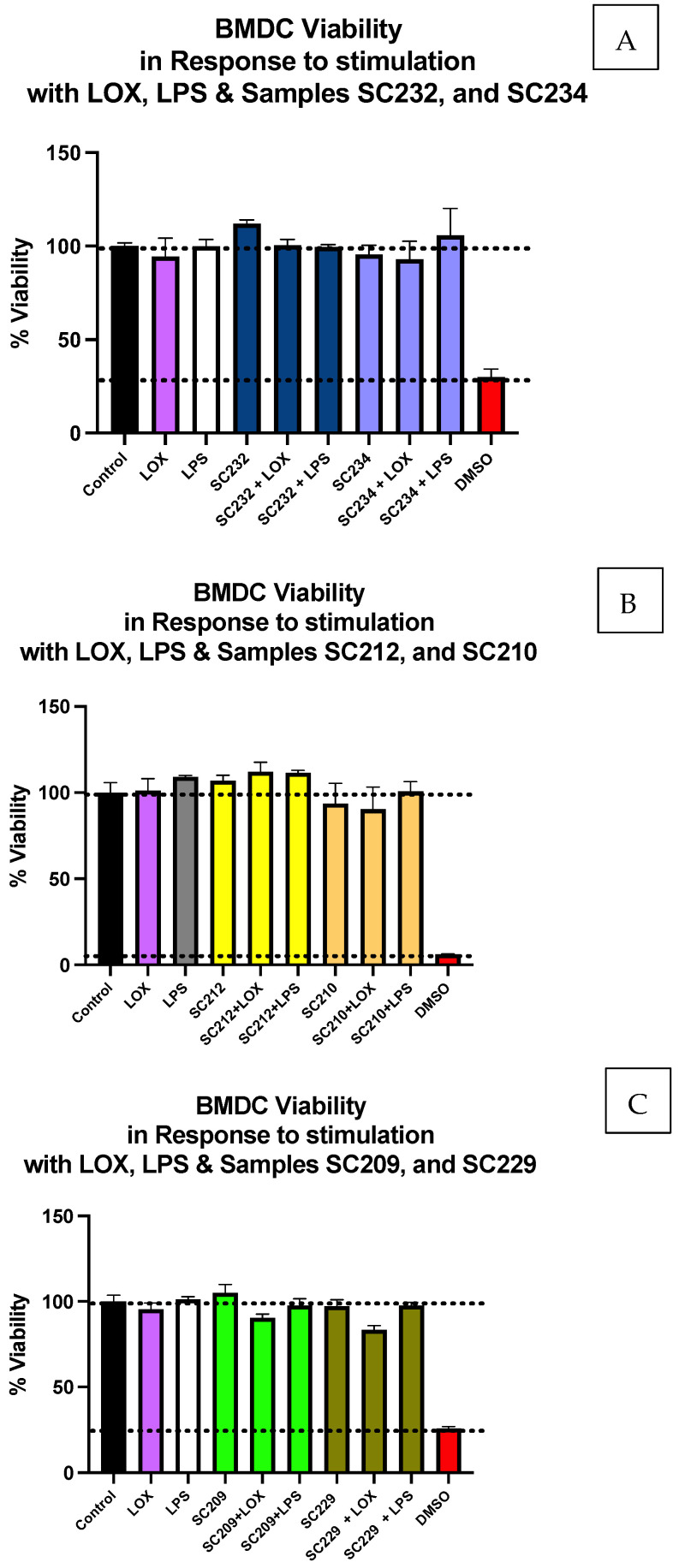
Exposure of LOX- and LPS-activated BMDCs to 25 mg/mL fermentates does not affect cell viability. BMDCs were seeded at 1 × 10^6^ cells/mL and incubated overnight at 37 °C in 5% CO_2_. The following day, the cells were stimulated with 25 mg/mL of the following fermentates: (**A**) SC212 and SC210; (**B**) SC232 and SC234; or (**C**) null SC209 and SC229. These were incubated for 1 h at 37 °C in 5% CO_2_ and subsequently exposed to LOX 0.5 mM and LPS 100 ng/mL before being incubated overnight under the same conditions. After 24 h, the cells were incubated with MTS CellTiter Aqueous One Solution for 3 h at 37 °C in 5% CO_2_. Viability is represented as a percentage, comparing the samples to viable untreated control cells. Data are presented as the mean ± SEM of three replicates.

**Figure 2 foods-13-02392-f002:**
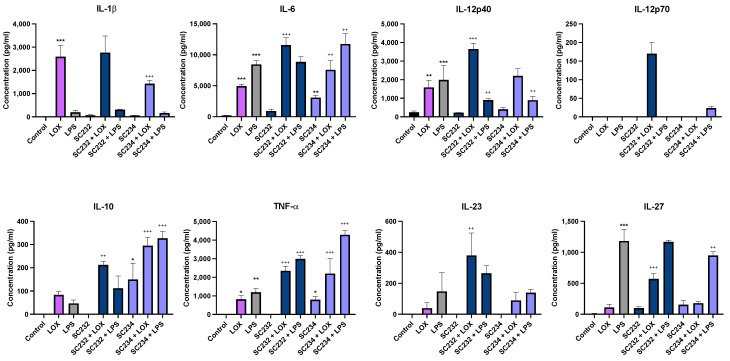
Exposure of LOX- and LPS-activated BMDCs to 25 mg/mL fermentates results in the secretion of IL-1β, IL-6, IL-10, TNF-α, IL-12p40, IL-12p70, IL-23, and IL-27. BMDCs were seeded at 1 × 10^6^ cells/mL and incubated overnight at 37 °C in 5% CO_2_. The following day, the cells were stimulated with 25 mg/mL fermentate, incubated for 1 h at 37 °C in 5% CO_2_, and subsequently exposed to LOX 0.5 mM and LPS 100 ng/mL before being incubated overnight under the same conditions. Supernatants were removed after 24 h and an ELISA was performed for IL-1β, IL-6, IL-10, TNF-α, IL-12p40, IL-12p70, IL-23, and IL-27. Data are presented as the mean ± SEM of three replicates. Significance was determined using a one-way ANOVA with a Newman–Keuls post-test. Output *p*-value, APA style where: 0.033 (*), 0.002 (**), and <0.001 (***); and each symbol is explained as (1) “*” represents comparing the control cells to the LOX or LPS or fermentate alone, and (2) “+” represents comparing the LOX or LPS to the fermentates with LOX or LPS. Clarified different symbols mean different explanations but both follow the APA style.

**Figure 3 foods-13-02392-f003:**
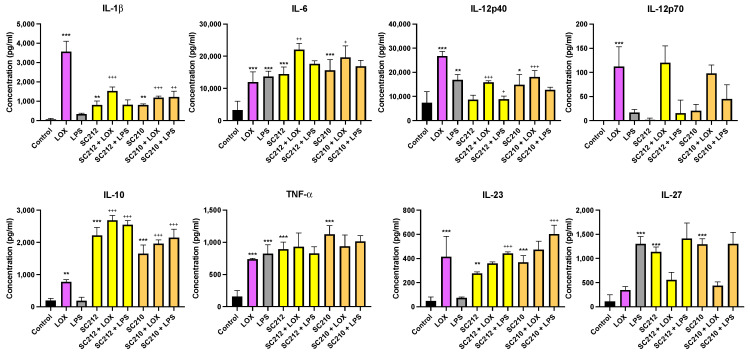
Exposure of LOX- and LPS-activated BMDCs to 25 mg/mL fermentates results in the secretion of IL-1β, IL-6, IL-10, TNF-α, IL-12p40, IL-12p70, IL-23, and IL-27. BMDCs were seeded at 1 × 10^6^ cells/mL and incubated overnight at 37 °C in 5% CO_2_. The following day, the cells were stimulated with 25 mg/mL fermentate, incubated for 1 h at 37 °C in 5% CO_2_, and subsequently exposed to LOX 0.5 mM and LPS 100 ng/mL before being incubated overnight under the same conditions. Supernatants were removed after 24 h and an ELISA was performed for IL-1β, IL-6, IL-10, TNF-α, IL-12p40, IL-12p70, IL-23, and IL-27. Data are presented as the mean ± SEM of three replicates. Significance was determined using a one-way ANOVA with a Newman–Keuls post-test. Output *p*-value, APA style where: 0.033 (*), 0.002 (**), and <0.001 (***); and each symbol is explained as (1) “*” represents comparing the control cells to the LOX or LPS or fermentate alone, and (2) “+” represents comparing the LOX or LPS to the fermentates with LOX or LPS. Clarified different symbols mean different explanations but both follow the APA style.

**Figure 4 foods-13-02392-f004:**
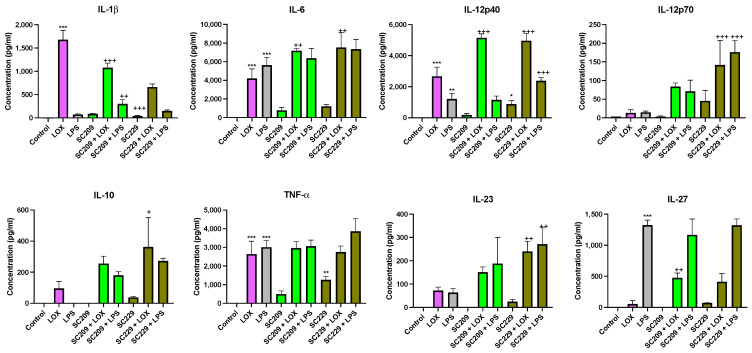
Exposure of LOX- and LPS-activated BMDCs to 25 mg/mL fermentates results in the secretion of IL-1β, IL-6, IL-10, TNF-α, IL-12p40, IL-12p70, IL-23, and IL-27. BMDCs were seeded at 1 × 10^6^ cells/mL and incubated overnight at 37 °C in 5% CO_2_. The following day, the cells were stimulated with 25 mg/mL fermentate, incubated for 1 h at 37 °C in 5% CO_2_, and subsequently exposed to LOX 0.5 mM and LPS 100 ng/mL before being incubated overnight under the same conditions. Supernatants were removed after 24 h and an ELISA was performed for IL-1β, IL-6, IL-10, TNF-α, IL-12p40, IL-12p70, IL-23, and IL-27. Data are presented as the mean ± SEM of three replicates. Significance was determined using a one-way ANOVA with a Newman–Keuls post-test. Output *p*-value, APA style where: 0.033 (*), 0.002 (**), and <0.001 (***); and each symbol is explained as (1) “*” represents comparing the control cells to the LOX or LPS or fermentate alone, and (2) “+” represents comparing the LOX or LPS to the fermentates with LOX or LPS. Clarified different symbols mean different explanations but both follow the APA style.

**Figure 5 foods-13-02392-f005:**
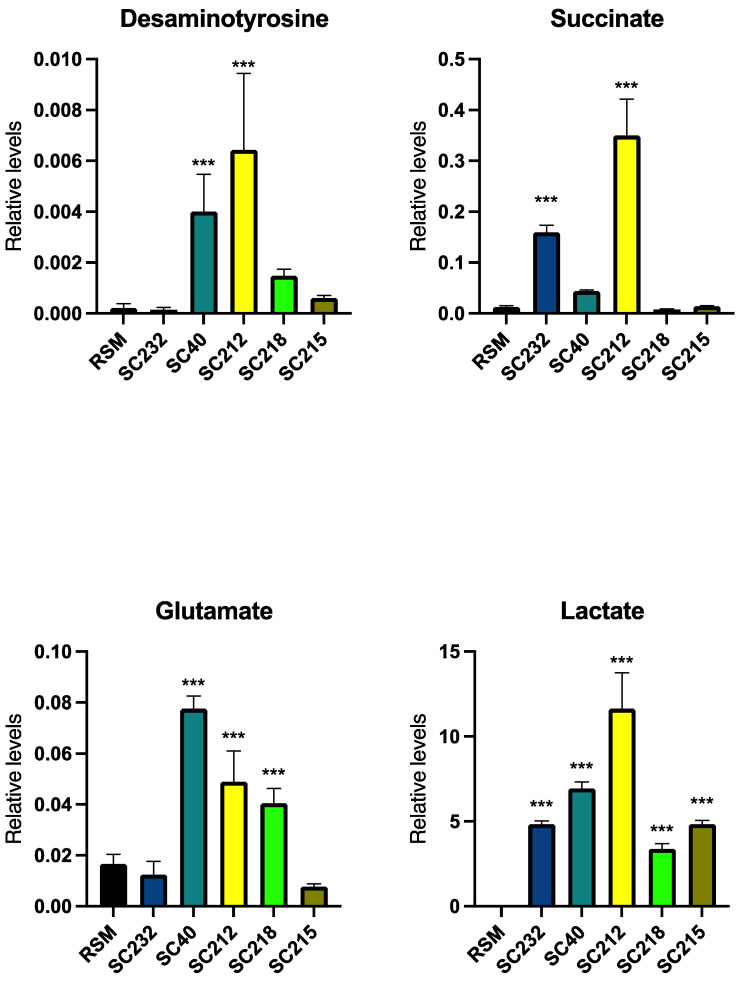
Metabolite levels in fermentates, illustrated using GraphPad Prism V10. Data are presented as the mean ± SEM of six replicates. Significance was determined using a one-way ANOVA with a Newman–Keuls post-test. Output *p*-value, APA style: 0.033 (*), 0.002 (**), and <0.001 (***), comparing the fermentate samples to the RSM.

**Table 1 foods-13-02392-t001:** Summary of the differing effects on cytokine secretion in the presence of SC232, SC234, SC212, and SC210 in LOX- or LPS-activated BMDCs. Green indicates increased cytokine secretion and red indicates decreased cytokine secretion. Significance was determined using a one-way ANOVA with a Newman–Keuls post-test. Output *p*-value, APA style: 0.033 (*), 0.002 (**), and <0.001 (***); “+” represents comparing the LOX or LPS to the fermentate with LOX or LPS.

	SC232	SC234	SC212	SC210	
+LOX		***	***	***	IL-1β
	***	**	**	*	IL-6
	**	***	***	***	IL-10
	***	***			TNF-α
	***		***	***	IL-12p40
					IL-12p70
	**				IL-23
	**				IL-27
+LPS		**		**	IL-1β
					IL-6
		***	***	***	IL-10
	***	***			TNF-α
	**	**	*		IL-12p40
					IL-12p70
			***	***	IL-23
		**			IL-27

## Data Availability

The raw data supporting the conclusions of this article will be made available by the authors on request.

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
