# Peer review of "Novel Dairy Fermentates Have Differential Effects on Key Immune Responses Associated with Viral Immunity and Inflammation in Dendritic Cells"

_foods, 2024, doi:10.3390/foods13152392_

Round 1
Reviewer 1 Report
Comments and Suggestions for Authors
This study has some significance and valuable. The manuscript was well written. However, there are some statement or questions need to be modified or responded before the article be considered for publication in “Foods”:
1. In the Introduction and Discussion sections, the space between paragraphs is too much, and there are even redundant empty lines, such as Line 86.
2. Lines 98-104, in the Introduction section, only need to tell the readers why you did this study, there is no need to show the results and data. Thus, these sentences are recommended to delete or reorganize.
3. Line 109 and 116, the treatments of “autoclaved” and ”heat-treated” should be provided the specific conditions, including temperature and time.
4. Line 136: "treated with 25 mg/mL of the fermentate", does 25 mg/mL here mean adding 25 mg fermentate per ml of cell culture medium? In this case, the fermentate needs to be dried first, but the authors did not mention how to dry the fermentate in the method. In addition, the authors did not mention how to prepare the sample after the fermentation is finished, such as whether centrifugation is required? Is homogenization required? However, in the metabolomics analysis, the samples were centrifuged at 13500 x g for 15 minutes, and then the supernatants were used to identify. Under this centrifugation condition, the bacterial cells and some macromolecular substances have been removed. So, is the fermentate used here (to treat the cells) also the supernatant after 15 minutes of centrifugation at 13500 x g? If the samples used for metabolomic identification do not the same as those used for cell treatment, the metabolomic data will not be convincing.
5. Lines 164-167, in the Method section, only need to tell the reader how you did it, and also do not need to show the results and data. Therefore, this sentence is suggested to be deleted.
6. In Figure 1, under the same treatment conditions, why was the BMDC Viability of DMSO group in picture B much lower than that in picture A and C?
7. In Section 3.5, why did the metabolomics analysis only detect 45 metabolites? In a metabolomics analysis, this identification amount appears to be much small. In addition, the authors only provided the differences between desaminotyrosine, succinate, glutamate, and lactate. Are there no any significant differences in the other metabolites? Is it possible that other differential metabolites also play an important role? The contents of all the 45 metabolites in different samples should be provided in the Supplementary Tables.
Reviewer 2 Report
Comments and Suggestions for Authors
Introduction:
Line 66: it is not the first study to explore the inflammatory effect of those fermentates. Previously, a published paper has used the same fermentates. Please revise the following: (Finnegan, D.; Mechoud, M.A.; FitzGerald, J.A.; Beresford, T.;Mathur, H.; Cotter, P.D.; Loscher, C.Novel Fermentates Can Enhance Key Immune Responses Associated with Viral Immunity. Nutrients 2024, 16,1212. https://doi.org/10.3390/nu16081212.
From Line 87to 89: I think it might more appropriate for the authors to locate those statements in the below paragraph.
Line 134: The authors mentioned macrophage cells instead of BMDC. Please revise.
Line 138: Is there any reference that support the LPS dosage on BMDC?. Was this dosge used previously to induce inflammation in BMDC?
Line 144: Is it possible for the authors to present the confidence interval for the ELISA in supplementary materials?
Also, indicating the type of used ELISA is required (direct, SANDWICH, etc,)
Line 167: Which software has been used to determine the statistical analysis?.
Please indicate how many times the experiements were performed.
Line 174: the ethical statement should be included in the manuscript according to the journal guidelines. Please revise
Result Section:
Figure 1: The authors need to explain each label (A) (B) (C). I recommend also to mention the source of bacteria for each fermentates to facilitate the understanding.
Figure 1 B, SC212 and SC210 is reptead three times?
Figure 1C , SC209 is also repeated four times. what is the difference between them.
The authors should consider to createa easier illustration of the results. Perhaps using table is more suitable.
Please indicate the statistical difference also between the groups.
In Figure 3, it has been claimed that SC212 and SC210 has anti-inflammatory properties (mentioned in conlusion), however, there are few points appeared in the results which are controversial. As apperead in the figure, cells treated with SC212 alone has increased the production of IL-1 beta, IL-6, TNF-alpha and IL-23.
The same points can be indicated to SC210.
Line 293: I think that presenting the H1-NMR spectrum results would be recommended.
Line 325 to 328: Please revise this statement since it has been copied from a previous research
Comments on the Quality of English Language
The English language, as recommended, need to be revised accordingly.
Round 2
Reviewer 1 Report
Comments and Suggestions for Authors
The manuscript was well modified and improved.
Reviewer 2 Report
Comments and Suggestions for Authors
The authors have addressed the comments and changes have been made in the manuscript accordingly